# Redirecting immune signaling with cytokine adaptors

Gita C. Abhiraman[1,2], Karsten D. Householder [1,2], Grayson E. Rodriguez [1,2], Caleb R. Glassman [1], Robert A. Saxton[1], Cort B. Breuer [2,3], Steven C. Wilson[1], Leon Su [1], Michelle Yen[1], Cynthia Hsu[4], Venu G. Pillarisetty [4], Nathan E. Reticker-Flynn [3] & K. Christopher Garcia [1,5] ✉

Cytokines are signaling molecules that coordinate complex immune processes and are frequently dysregulated in disease. While cytokine blockade has become a common therapeutic modality, cytokine agonism has had limited utility due to the widespread expression of cytokine receptors with pleiotropic effects. To overcome this limitation, we devise an approach to engineer molecular switches, termed cytokine adaptors, that transform one cytokine signal into an alternative signal with a different functional output. Endogenous cytokines act to nucleate the adaptors, converting the cytokine–adaptor complex into a surrogate agonist for a different cytokine pathway. In this way, cytokine adaptors, which have no intrinsic agonist activity, can function as conditional, context-dependent agonists. We develop cytokine adaptors that convert IL-10 or TGF-β into IL-2 receptor agonists to reverse T cell suppression. We also convert the pro-inflammatory cytokines IL-23 or IL-17 into immuno-suppressive IL-10 receptor agonists. Thus, we show that cytokine adaptors can convert immunosuppressive cytokines into immunostimulatory cytokines, or vice versa. Unlike other methods of immune conversion that require cell engineering, cytokine adaptors are soluble molecules that leverage endogenous cues from the microenvironment to drive context-specific signaling.

Cytokines act locally to activate or suppress immune cells at the site of infection or disease. The balance between stimulatory and suppressive cytokines in a diseased tissue can ultimately determine patient outcomes. Immunosuppressive cytokines, such as TGF-β and IL-10, are overexpressed in the local tumor microenvironment and associated with poor clinical outcomes[1,2]. Conversely, pro-inflammatory cytokines are overexpressed in several autoimmune diseases and drive pathological inflammation[3,4]. IL-23 and IL-17 are upregulated in animal models of colitis and in the colons of patients with inflammatory bowel disease[5–9]. An increasing number of cytokine antagonist therapies have been clinically approved, with the goal of blocking these deleterious immune responses[10,11]. IL-23, IL-17, and TNF-α inhibitors are approved for the treatment of psoriasis, inflammatory bowel disease, and rheumatoid arthritis[4]. However, systemic immunosuppression can lead to adverse effects, including the reactivation of tuberculosis, hepatitis, or herpes zoster virus[12–14]. Furthermore, rates of relapse after discontinuation of cytokine antagonist therapy are high[15,16], suggesting that cytokine antagonism on its own may be insufficient in achieving lasting immune tolerance.

Cytokine receptor agonism has been less successful as a clinical strategy, with noted exceptions such as erythropoietin and granulocyte-macrophage colony-stimulating factor[17–20]. A major

[1]Department of Molecular and Cellular Physiology, Stanford University School of Medicine, 279 Campus Drive, Stanford, CA 94305, USA. [2]Program in Immunology, Stanford University School of Medicine, Stanford, CA 94305, USA. [3]Department of Otolaryngology – Head & Neck Surgery, Stanford University School of Medicine, Stanford, CA 94305, USA. [4]Department of Surgery, University of Washington School of Medicine, Seattle, WA 98112, USA. [5]Howard Hughes Medical Institute, Stanford University, Stanford, CA 94305, USA. ✉e-mail: kcgarcia@stanford.edu

limitation of cytokine agonist therapies is the need to administer cytokines systemically, leading to off-target toxicities caused by cytokine pleiotropy and widespread receptor expression[19,21]. For example, systemic IL-2 administration is associated with capillary leak syndrome, resulting in damage to the kidneys, heart, and brain[22,23]. Previous approaches to limit cytokine-associated toxicities include targeting via antibody fusion[24,25], protease-cleavable linkers[26,27], pH-selective muteins[28], and partial agonists with biased agonism[18,29,30].

We report here a strategy that leverages the specificity of cytokine antagonists to deliver a local cytokine agonist signal by using the target cytokine as a clustering scaffold. We engineer cytokine adaptors to induce context-dependent activation of target cytokine receptors based on the simple principle of receptor dimerization (Fig. 1a, b). Adaptor molecules are comprised of modular protein units, such as antibody components and cytokine receptor domains, and are engineered to simultaneously block one cytokine while inducing activation of an alternate cytokine pathway. They induce a context-dependent signal, with the potential to increase specificity local to the tumor or diseased tissue and reduce off-target cytokine signaling. We present cytokine adaptors that convert immunosuppressive signals to immunostimulatory signals and vice versa. Specifically, we engineer TGF-β-gated IL-2 (TGF-β→IL-2) and IL-10-gated IL-2 (IL-10→IL-2) cytokine adaptors that reverse T cell inhibition and have potential applications in cancer therapy. In addition, we engineer IL-23-gated IL-10 (IL-23→IL-10) and IL-17-gated IL-10 (IL-17→IL-10) cytokine adaptors with reverse function, converting IL-23 or IL-17 into IL-10 to suppress inflammatory cytokine production.

## Results

### Concept and design of cytokine adaptors
Our design goal was to engineer soluble adaptor molecules that would convert one signaling input into a new signaling output based on induced proximity (Fig. 1a). Native cytokines induce signaling through receptor dimerization. Based on this dimerization paradigm, we conceived of molecules that could convert an inhibitory cytokine into a stimulatory signal by: (1) blocking the inhibitory cytokine from binding to its receptor at two sites, and (2) leveraging the two-site binding of the antagonist to compel dimerization and activation of the stimulatory cytokine receptors via adaptors (Fig. 1b). This design incorporates binding domains targeting an input cytokine, such as single-chain variable fragments (scFv), single domain variable fragments from heavy chain-only antibodies (VHH), or soluble cytokine receptors themselves, linked to binding domains for the output cytokine receptors such as scFv, VHH, or dominant-negative cytokine mutants. This approach is generalizable to any input cytokine that can be bound by two binding modules, and any output cytokine that signals through a dimeric receptor.

### TGF-β→IL-2 adaptor induces pSTAT5 signaling in the presence of TGF-β
We sought to engineer molecules that could reverse immunosuppression in the context of cancer, for example, by inhibiting tumor-derived TGF-β and stimulating the IL-2 pathway. We built a model of such a molecule using structures of TGF-β and the scFv antagonist GC1008 (fresolimumab)[31–33], and of VHH binders to IL-2Rβ and γc[34] (Fig. 1c). TGF-β is a homodimer, to which two copies of scFv GC1008 can bind. We first engineered a two-component adaptor system in which one copy of GC1008 was linked to an N-terminal VHH against IL-2Rβ, IL-2RβNb6, and a second molecule in which GC1008 was linked to γcNb6, a VHH against γc (Fig. 1d, Supplementary Fig. 1a). We hypothesized that when both molecules were bound to TGF-β, they would compel dimerization of the IL-2Rβ and γc receptors, leading to downstream JAK/STAT activation akin to endogenous IL-2 signaling. To test the signaling capacity of the TGF-β→IL-2 cytokine adaptors, we stimulated YT-1 cells, an IL-2 responsive human NK cell line, with TGF-β

and equimolar TGF-β→IL-2 cytokine adaptor in a dose–response experiment. The combination of TGF-β with TGF-β→IL-2 adaptors induced 57% pSTAT5 $E_{max}$ in YT-1 cells compared to IL-2 (Fig. 1e, f, Supplementary Table 1). In human primary T cells, the TGF-β→IL-2 adaptors also induced a dose-dependent pSTAT5 response in the presence of TGF-β (Fig. 2e, Supplementary Table 2). TGF-β or the TGF-β→IL-2 adaptors on their own did not stimulate phospho-STAT5 (pSTAT5), demonstrating that cytokine adaptor signaling is dependent on the presence of TGF-β (Fig. 1e, f).

### Single-chain TGF-β→IL-2 cytokine adaptor reverses T cell inhibition and promotes cytotoxicity in primary human T cells
Though the two-component adaptor molecules demonstrated IL-2R agonism conditional upon TGF-β, this design was limited by its reduced $E_{max}$. Since IL-2 signaling is mediated by a heterodimer of IL-2Rβ/γc, two-component adaptors can potentially induce non-productive pairings of γc/γc and IL-2Rβ/IL-2Rβ in addition to signaling competent IL-2Rβ/γc dimers (Fig. 2a). To mitigate this issue, we engineered a single-component TGF-β→IL-2 adaptor by introduction of a long, unstructured linker between individual adaptor proteins (Fig. 2a). Only in the presence of TGF-β does adaptor binding enforce close enough proximity of receptor subunits to drive downstream signaling. We conducted structure-activity relationship studies by testing a series of molecules with varied linker lengths and orientations (Fig. 2b, Supplementary Fig. 1c–f). We simultaneously measured the ability of various constructs to inhibit pSMAD2/pSMAD3 signaling downstream of TGF-β and to induce pSTAT5 in activated primary human T cells. We found that TGF-β inhibition was improved in the N to C orientation of the IL-2Rβ and γc domains, and that the specificity of signaling was linker-length dependent (Fig. 2c, d, Supplementary Fig. 1c–f). We selected Adaptor T.3, which contained a 20 amino acid linker, for downstream studies, since it was more specific to the presence of TGF-β. In dose–response experiments in human primary CD4+ and CD8+ T cells, single-chain Adaptor T.3 signaled with an improved $E_{max}$ of 87 or 98% and $EC_{50}$ of 810 or 900pM in CD4+ and CD8+ T cells, respectively, compared to the two-component adaptor pair T.1 and T.2 (Fig. 2e, f, Supplementary Fig. 1c, Supplementary Table 2).

We then tested the functional effects of the optimized single-chain TGF-β→IL-2 adaptor on human T cells. Several cancer types are known to overexpress TGF-β, which contributes to immune evasion via its suppressive and antiproliferative effects on T cells[1,35,36]. TGF-β suppresses T cell receptor signaling, cytotoxicity, and proliferation, in contrast to the effects of IL-2[36,37] (Fig. 2g). Indeed, TGF-β suppressed proliferation of primary human CD4+ and CD8+ T cells in vitro over the course of six days (Fig. 2h, Supplementary Fig. 2a). However, addition of the TGF-β→IL-2 adaptor reversed the effects of TGF-β and promoted T cell proliferation in a TGF-β-dependent manner, to a similar extent as IL-2 treatment alone (Fig. 2h). TGF-β blockade with the anti-TGF-β scFv alone was not sufficient to promote T cell proliferation (Supplementary Fig. 2b). Moreover, treatment with TGF-β→IL-2 adaptor T.3 in the presence of TGF-β induced robust production of TNF-α and IFN-γ in primary T cells, indicating strong activation of IL-2R-pSTAT5 signaling (Fig. 2i, j, Supplementary Fig. 2c). The TGF-β→IL-2 adaptor T.3 also induced upregulation of CD25 (IL-2Rα), a maker of T cell activation, on both CD4+ and CD8+ T cells (Supplementary Fig. 2d). These results suggest that the TGF-β→IL-2 cytokine adaptor can reverse TGF-β mediated suppression and promote T cell effector functions.

### IL-10→IL-2 adaptors induce pSTAT5 signaling in the presence of IL-10
We next pursued the design of an IL-10→IL-2 cytokine adaptor since IL-10 is also dysregulated in several cancers and is associated with worse outcomes in cancer patients[38,39]. IL-10 decreases antigen presentation

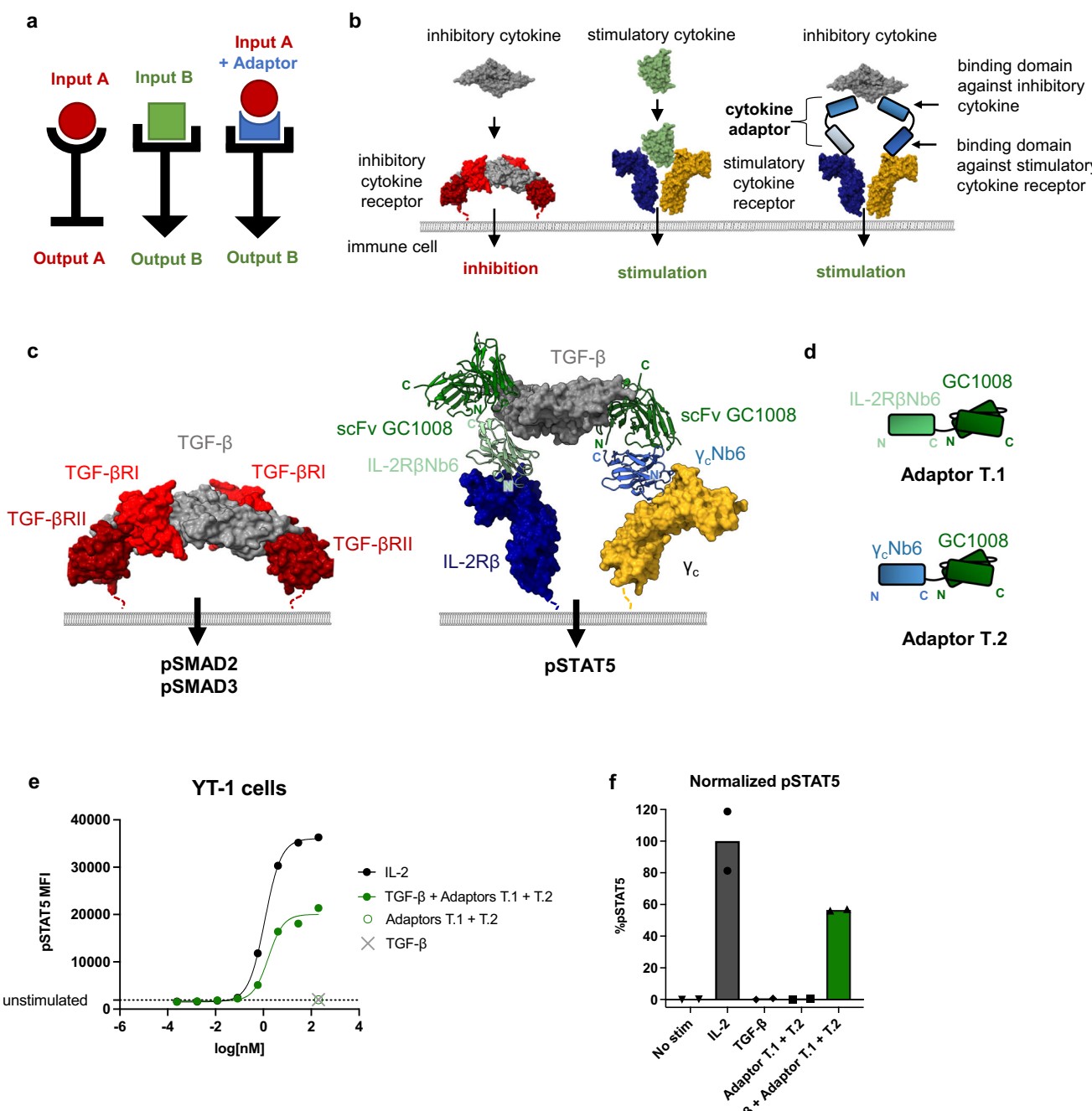

**Fig. 1 | TGF-β→IL-2 cytokine adaptors compel IL-2 receptor signaling in the presence of TGF-β. a** Conceptual overview of an adaptor. **b** Schematic illustrating cytokine adaptors, which convert an inhibitory cytokine into a stimulatory cytokine by blocking the inhibitory cytokine from binding to its receptor and instead compelling dimerization of the stimulatory receptor. Model is based on IL-2R (PDB ID: 2B5I) and TGF-βR (PDB ID: 2PJY). **c** Model comparing natural TGF-β signaling through TGF-βRI and TGF-βRII (left) vs. signaling through TGF-β→IL-2 cytokine adaptors (right) which dimerize IL-2Rβ and gamma-c (γ$_c$). Adaptor molecules are comprised of scFv GC1008 linked to a VHH against IL-2Rβ (IL-2RβNb6) or a VHH against γ$_c$ (γ$_c$Nb6). **d** Cartoon representations of TGF-β Adaptor T.1 and Adaptor T.2. **e, f** Adaptors T.1 and T.2 signal through pSTAT5 in the presence of TGF-β. **e** Dose–response curves for phospho-STAT5 in YT-1 cells stimulated for 20 min with human IL-2 or TGF-β with equimolar Adaptor T.1 and Adaptor T.2. Data plotted as the mean of $n = 2$ technical replicates. Data are representative of $N = 3$ independent experiments ($n = 2$, $N = 3$). **f** Phospho-STAT5 E$_{max}$ calculated from dose–response curves in YT-1 cells normalized to IL-2 E$_{max}$. Bar graphs represent mean, $n = 2$, $N = 3$. Source data are provided as a Source data file.

by dendritic cells by downregulating costimulatory molecules including CD86 and MHC class II[40,41]. The immunosuppressive effects of IL-10 on myeloid cells leads to the indirect suppression of T cells[30]. To convert IL-10 into an IL-2 agonist, we designed a pair of IL-10 adaptors by linking an inhibitory αIL-10 scFv 9D7[42] to IL-2RβNb6, and γcNb6 to IL-10Rα ectodomain (Fig. 3a, b, Supplementary Fig. 3a). In YT-1 cells,

the IL-10→IL-2 adaptors induced a robust dose-dependent pSTAT5 signal in the presence of IL-10, akin to the pSTAT5 response induced by IL-2 (Fig. 3c, d, Supplementary Table 3). In human primary T cells, the addition of increasing concentrations of IL-10→IL-2 adaptor in the presence of a fixed concentration of IL-10 resulted in pSTAT3 inhibition and pSTAT5 activation (Fig. 3e).

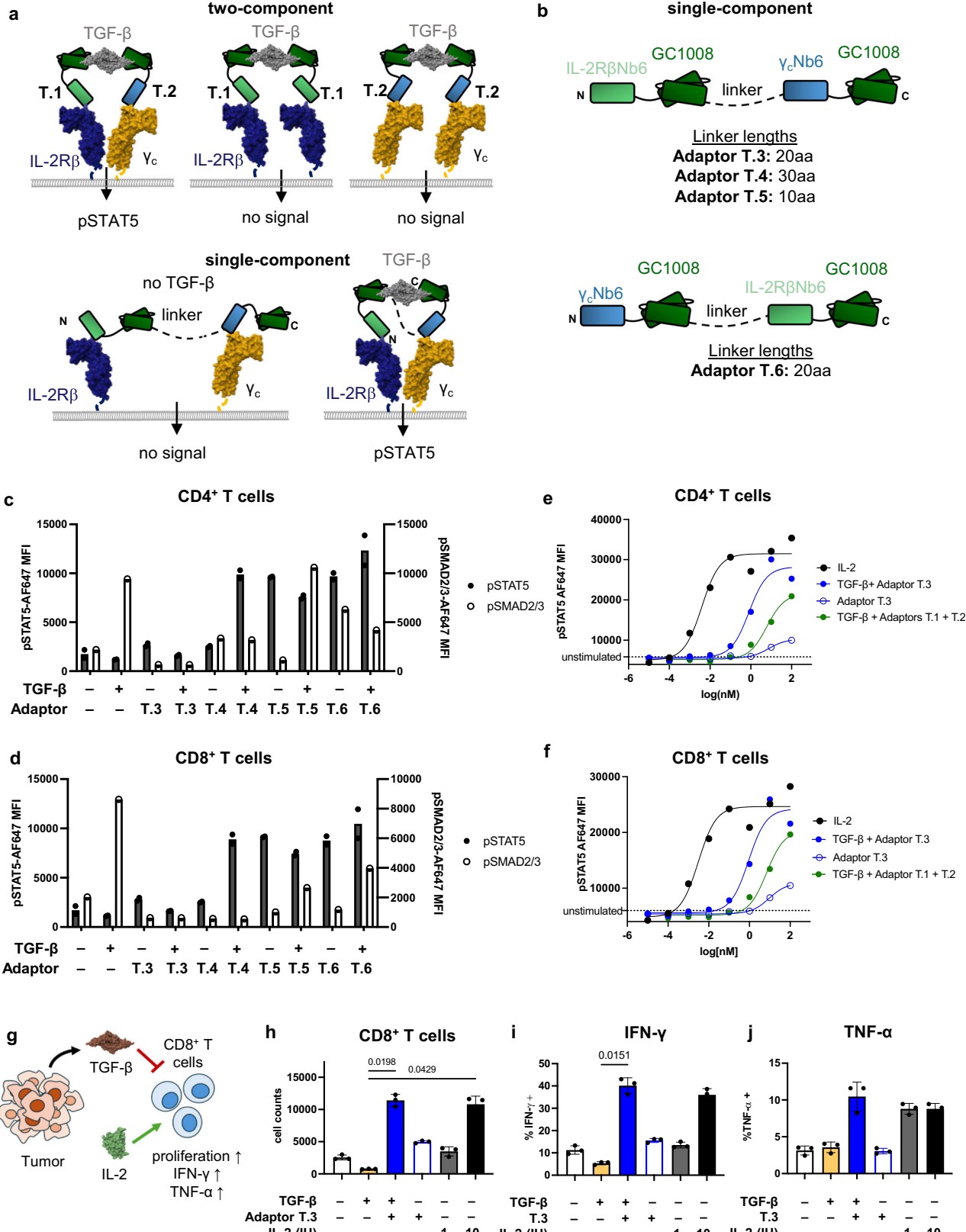

## IL-23→IL-10 and IL-17→IL-10 adaptors suppress LPS-induced cytokine production

We next designed cytokine adaptors to convert an immunostimulatory cytokine into an immunosuppressive cytokine, for potential applications in autoimmune disease. The advantage of such an approach could be that restricting the immunosuppressive signal to the inflammatory lesion could reduce off target effects. We targeted IL-23 and IL-17, which are upregulated in psoriasis, inflammatory bowel disease, and rheumatoid arthritis[43]. Inhibitors of IL-23 are clinically approved for the treatment of psoriasis and inflammatory bowel disease[43,44], and inhibitors of IL-17A are used in the treatment of plaque psoriasis, psoriatic arthritis, and ankylosing spondylitis[45]. We chose to

**Fig. 2 | Single-component cytokine adaptor T.3 converts TGF-β stimulation to IL-2 signaling in human primary T cells. a** Schematic illustrating rationale for designing single-component cytokine adaptors. In the two-component system (top), non-productive pairs of T.1/T.1 and T.2/T.2 are not able to dimerize IL-2Rβ and γ_c. To engineer a single-component adaptor (bottom), the two components are linked by a flexible linker, such that receptor dimerization is compelled only in the presence of TGF-β. **b** Several single-component cytokine adaptors were designed with varying orientations and linker lengths. Adaptors T.3, T.4, and T.5 contain N-terminal IL-2RβNb6 and linker lengths of 20aa, 30aa, or 10aa respectively, while Adaptor T.6 features a reversed orientation. **c, d** Single-component adaptors suppress TGF-β-induced pSMAD2/pSMAD3 signaling while stimulating pSTAT5 signaling in human CD4$^+$ (**c**) and CD8$^+$ (**d**) T cell blasts. Data is represented as mean fluorescence intensity of phospho-STAT5 ($n = 2$) and pSMAD2/pSMAD3

($n = 1$) of T cells treated with 10 nM TGF-β +/- Adaptor T.3, T.4, T.5, or T.6. Bar graphs represent mean ± SD. **e, f** Single-component adaptor T.3 improves dose–response signaling in human primary T cells. Dose–response curves are shown for phospho-STAT5 in human CD4$^+$ (**e**) and CD8$^+$ (**f**) T cell blasts stimulated for 20 min with human IL-2, Adaptor T.3, TGF-β with equimolar Adaptor T.3, or TGF-β with equimolar Adaptor T.1 and Adaptor T.2. Data are plotted as mean, $n = 2$, $N = 3$. **g** Schematic illustrating the suppressive effects of TGF-β and stimulatory effects of IL-2 on T cell proliferation and cytotoxicity in the tumor microenvironment. **h–j** Adaptor T.3 increases CD8$^+$ T cell proliferation (**h**), IFN-γ (**i**), and TNF-α (**j**) production in the presence of TGF-β. Bar graphs represent mean, $n = 3$, $N = 3$. * indicates $p < 0.05$, ** indicates $p < 0.01$ by one-way non-parametric ANOVA (Kruskal–Wallis test) with multiple comparisons. Bar graphs represent mean ± SD. Source data are provided as a Source data file.

convert IL-23 or IL-17 into an IL-10 signal, since IL-10 has important anti-inflammatory and tolerogenic activity, via inhibition of macrophage and dendritic cell functions such as antigen presentation and inflammatory cytokine production[46,47]. IL-10 receptor agonists have been investigated in clinical trials for the treatment of Crohn's disease, psoriasis, and rheumatoid arthritis, but have been hampered by toxicity from off-target effects[48–50].

We generated IL-23→IL-10 adaptors by fusing VHH 37D5, which binds to the IL-23p19 subunit, to a VHH against IL-10Rβ (Nb2). We fused VHH 22E11, which binds to the IL-23p40 subunit, with a dominant negative monomeric mutant of IL-10, which binds to IL10Rα, but is not signaling competent due to weakened IL-10Rβ affinity (Fig. 4a–c, Supplementary Fig. 3b). We tested several configurations of IL-23→IL-10 adaptor pairs in THP-1 cells, a human monocytic cell line that is responsive to wild-type IL-10 and a monomeric form of IL-10 (M) but not to IL-23 (Fig. 4c, d). We identified a pair of IL-23→IL-10 adaptors that induced a dose-dependent pSTAT3 response in the presence of IL-23 with an EC$_{50}$ of 24pM (Fig. 4c, d, Supplementary Table 4).

To test the functional properties of the IL-23→IL-10 adaptors, we stimulated human peripheral blood mononuclear cells (PBMCs) with bacterial lipopolysaccharides (LPS) alone or in the presence of IL-23 and/or IL-23→IL-10 adaptors. Stimulation of human PBMCs with LPS promotes production of the inflammatory cytokines TNF-α, IL-6, and IL-1β, which is reversed by the addition of IL-10 in culture (Fig. 4e–g). Critically, the IL-23→IL-10 adaptors suppressed LPS-induced production of TNF-α, IL-6, and IL-1β only in the presence of IL-23, and to a similar extent as IL-10 treatment alone (Fig. 4e–g). This shows that the IL-23→IL-10 adaptors suppress inflammatory cytokine production in human primary cells in an IL-23 dependent manner.

Using the same principle, we designed IL-17→IL-10 adaptors with the goal of blocking endogenous IL-17A signaling through IL-17RA and IL-17RC and instead compelling dimerization of IL-10Rα and IL-10Rβ (Fig. 5a)[51,52]. We generated the first IL-17→IL-10 adaptor by linking a VHH binding to IL-10Rβ (Nb2) with VHH-76, a fragment of the IL-17A binding antibody netakimab, with a 10 amino acid Gly-Ser linker (Fig. 5b). A second adaptor molecule was generated by fusing the IL-10 dominant negative mutant (IL-10DN) to an scFv fragment of the IL-17A binding human antibody, CAT2200, with a 10 amino acid Gly-Ser linker (Fig. 5b)[53]. The pair of molecules, Adaptor 17.1 and Adaptor 17.2, induced a dose-dependent pSTAT3 response (Fig. 5c, d). We tested whether these molecules could recapitulate the functional effects of IL-10 in human PBMCs stimulated with LPS. Similar to the IL-23→IL-10 adaptors, the IL-17→IL-10 adaptors suppressed LPS-mediated production of TNF-α, IL-6, and IL-1β in the presence of IL-17A (Fig. 5e–g). The suppressive function of the adaptors was specific to the presence of IL-17A.

Taken together, these results show that IL-23→IL-10 and IL-17→IL-10 adaptors can induce signaling akin to IL-10, dependent on the presence of IL-23 or IL-17. Furthermore, the adaptor molecules

recapitulate IL-10 function in vitro, suppressing LPS-mediated production of inflammatory cytokines in human primary cells.

## Discussion

Cytokine adaptors are soluble factors that convert a dysregulated cytokine into a desired therapeutic activity, with the potential to be administered systemically. The only requirements to this design are that the antagonist bind at two sites on the target cytokine: either one site on a homodimeric cytokine like TFG-β or IL-10, or at two different sites for asymmetric cytokine targets like IL-23, to enable dimerization of the desired receptors. In this sense, the target cytokine is exploited as a locally expressed clustering scaffold to induce dimeric receptor signaling. Adaptors offer improved specificity compared to ordinary cytokine therapy, since they compel dimerization and signaling of the IL-10 or IL-2 receptors specifically in the presence of a target molecule (TGF-β, IL-10, IL-23, or IL-17) that is upregulated in disease. Given that the cytokine activity conversion will occur within a local milieu, adaptors could provide a unique strategy to bypass pleiotropy and minimize toxicity. Previous attempts to rewire local cytokine signaling have included the use of engineered switch receptors in the context of adoptive cell therapy. These attempts utilize chimeric receptors that bind to inhibitory molecules, such as TGF-β or PD-L1, and induce T cell activation via a CD28 intracellular domain[54–57]. For clinical translation, these methods require patient apheresis and cellular engineering.

By contrast, the cytokine adaptors that we present act as soluble switch receptors by signaling through endogenously expressed receptors. Unlike autologous cell products, cytokine adaptors could be used off the shelf, with the potential to be easily manufactured at a fraction of the cost. In structure-activity relationship studies, we demonstrate how the specificity and signaling strength of adapoer molecules can be further modulated by altering the orientation and linker length between binding modules. There are many different formatting modalities that could be implemented to create cytokine adaptors for various disease applications. In our adaptor design, we used pre-existing components at our disposal, but bespoke engineering could undoubtedly yield new adaptor combinations for further study.

A major goal of cancer immunotherapy is to increase immune cell infiltration and activation in tumors. Cytokine therapies, such as IFN-α and IL-2, have been leveraged to stimulate the antitumor immune response in certain cancer types[19,20]. However, a major limitation of these therapies is toxicity caused by systemic receptor activation[19,21]. Our TGF-β→IL-2 and IL-10→IL-2 adaptors have the potential to reduce off-target effects by inducing signaling preferentially where the target cytokine is overexpressed, such as the tumor microenvironment. In addition, they deliver two-in-one function in a dual ability to block the immunosuppressive effects of TGF-β or IL-10, in addition to conferring T cell stimulation and activation through the IL-2 pathway.

Conversely, we also demonstrate the potential for cytokine adaptors to be leveraged for the suppression of inflammatory

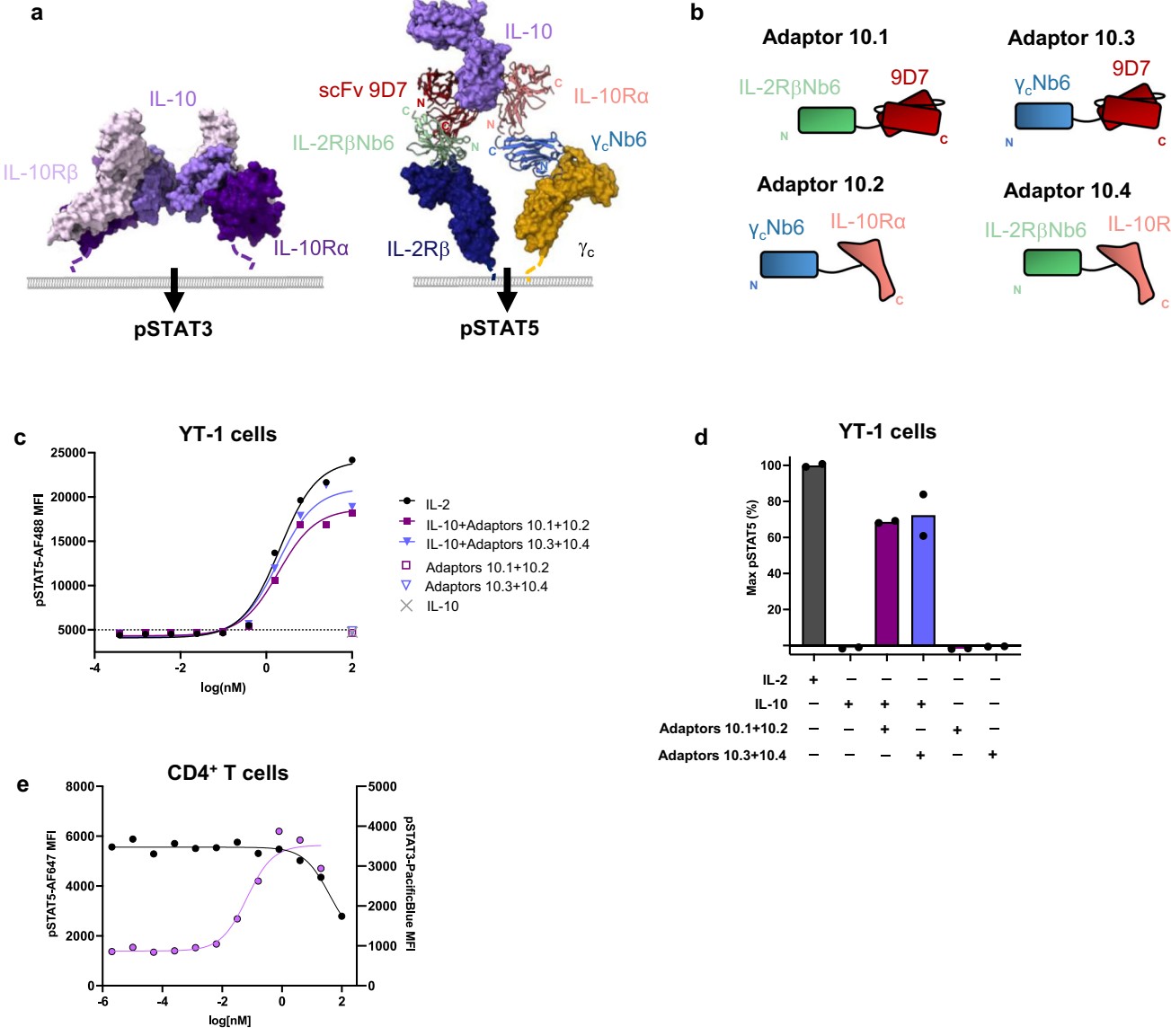

**Fig. 3 | IL-10→IL-2 adaptors convert IL-10 stimulation into IL-2 signaling.**
**a** Model comparing natural IL-10 signaling through IL-10Rα and IL-10Rβ (left, PDB ID: 6X93) vs. signaling through IL-10→IL-2 cytokine adaptors (right) which dimerize IL-2Rβ and γc. Adaptors are modeled using structures of IL-10 and scFv 9D7 (PDB ID: 1LK3) overlaid with IL-10 and IL-10Rα (PDB ID: 6X93), IL-2RβNb6 bound to IL-2Rβ (PDB ID: 7S2S) and γcNb6 bound to γc (PDB ID: 7S2R). **b** Cartoon representations of adaptors 10.1, 10.2, 10.3, and 10.4. **c** IL-10→IL-2 adaptors induce dose-dependent pSTAT5 signal similar to IL-2. Dose−response curves are shown for pSTAT5 in YT-1 cell line stimulated for 20 min with human IL-2 or IL-10 with equimolar Adaptor 10.1

and 10.2 or Adaptor 10.3 and 10.4. Data are plotted as mean, $n = 2$, $N = 3$.
**d** Normalized pSTAT5 response in YT-1 cells stimulated with the indicated cytokines or adaptors, normalized as a percentage of maximum IL-2 pSTAT5. Data are plotted as mean, $n = 2$. **e** Adaptors 10.1 and 10.2 suppress IL-10-induced pSTAT3 and promote IL-2 like pSTAT5 signaling. Dose−response curves for pSTAT5 and pSTAT3 are shown in human CD4⁺ T cell blasts stimulated with 10 nM IL-10 and increasing concentrations of equimolar Adaptor 10.1 and 10.2. $n = 2$ replicates per condition. Source data are provided as a Source data file.

cytokines. In autoimmune conditions including psoriasis, rheumatoid arthritis, and inflammatory bowel disease, IL-23 and IL-17 inhibitors have been clinically effective in ameliorating disease[45,58]. The potential of IL-10 agonism in the treatment of autoimmune disease has been explored but is limited due to off-target effects[48–50]. Our design of IL-23-gated and IL-17-gated IL-10 explore avenues for context-specific immunosuppression. Conversion of IL-23 or IL-17 to IL-10 with cytokine adaptors may have the potential to dampen inflammation downstream of myeloid activation and IFN-γ secretion, similar to the functions of IL-10.

In summary, we designed cytokine adaptors to enable immune conversion between a stimulatory and suppressive signal, and vice versa. Due to the modular nature of cytokine adaptor design, we

anticipate that adaptors can be readily developed for other cytokine pairs. Beyond cytokines, adaptor molecules can also be used to target other pathological markers, such as tumor markers or autoimmune antigens, and convert these into receptor agonists. We anticipate that this strategy will be broadly applicable to other receptor systems and disease contexts.

**Limitations of study**
In this study, we demonstrate proof of concept design of cytokine adaptors, which convert one cytokine signal into another. A limitation of our work is that our cytokine adaptors were engineered to be human-specific. Due to a lack of species cross-reactivity of cytokine and cytokine receptor binding components, we were unable to test our

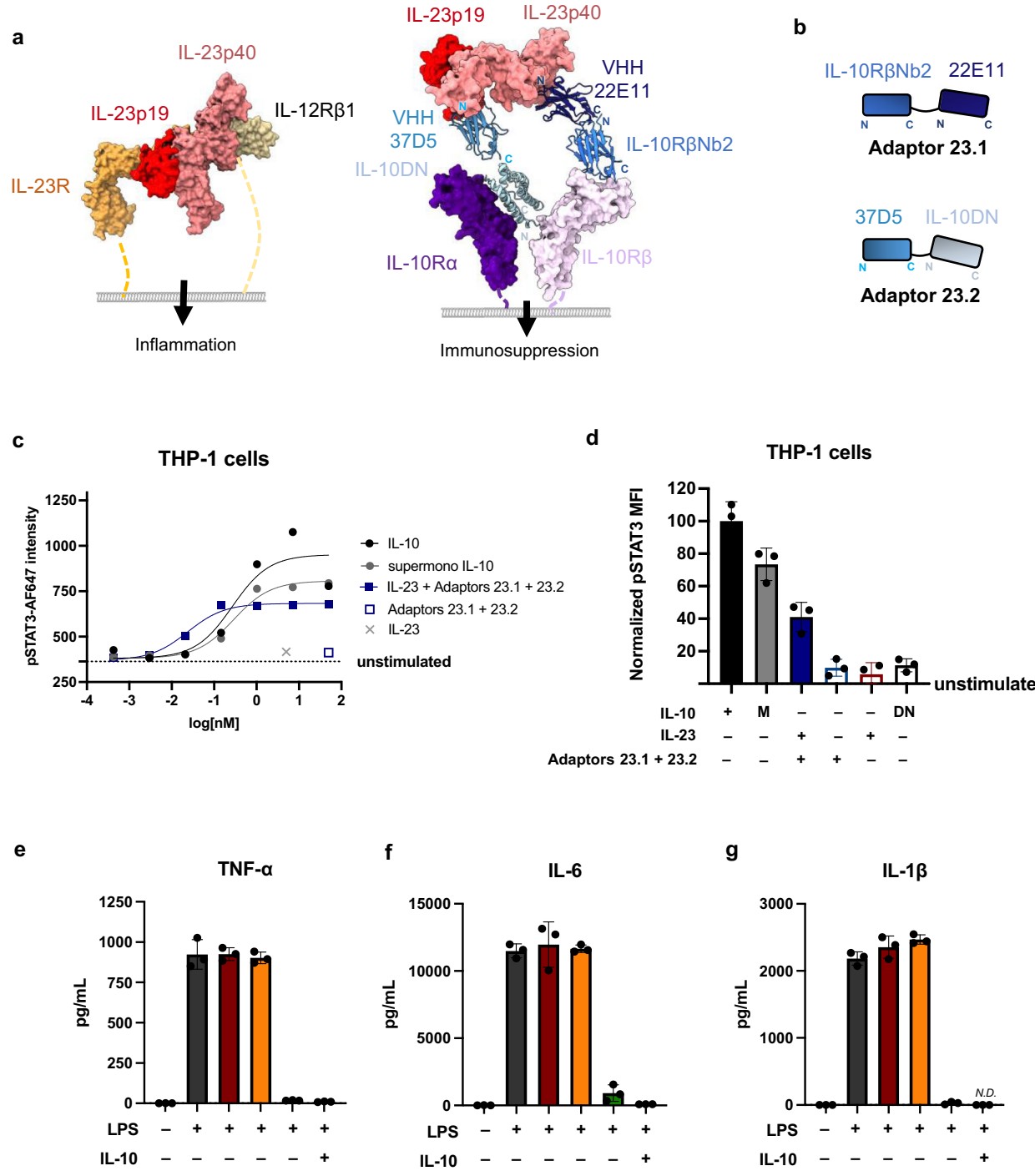

**Fig. 4 | IL-23→IL-10 cytokine adaptors mimic effects of IL-10. a** Model comparing natural IL-23 signaling through IL-23R and IL-12Rbβ1 (left, PDB ID: 6XDQ) vs. signaling through IL-23→IL-10 cytokine adaptors (right) which dimerize IL-10Rα and IL-10Rβ. Model of IL-23→IL-10 adaptors is comprised of IL-23p19 binder VHH 37D5 and IL-23p40 binder VHH 22E11 and IL-23p19 (PBD ID: 4GRW) overlaid with structures of IL-10 receptor complex (PDB ID: 6X93). **b** Cartoon representation of IL-23→IL-10 cytokine adaptor design. **c, d** IL-23→IL-10 adaptors induce a pSTAT3 signal in THP-1 cells similar to IL-10. **c** Dose–response curves for phospho-STAT3 in THP-1 cell line stimulated for 20 min with human IL-10, a supermonomeric version of IL-10, or IL-23 with equimolar IL-23→IL-10 cytokine adaptors. Data plotted as mean phospho-STAT3 mean fluorescence intensity, $n = 2$, $N = 3$. **d** Normalized pSTAT3 response in THP-1 cells stimulated with 10 nM WT IL-10, a supermonomeric variant of IL-10 (M), a monomeric dominant negative mutant of IL-10 (DN), IL-23, or IL-23 with the IL-23 → IL-10 adaptors. $n = 2$, $N = 3$. Data are plotted as mean ± SD. **e–g** IL-23→IL-10 adaptors suppress LPS-induced production of inflammatory cytokines. TNF-α (**e**), IL-6 (**f**), and IL-1β (**g**) quantified by ELISA in supernatants from human PBMCs stimulated for 24 h with LPS and 10 nM IL-10, IL-23, or IL-23 → IL-10 adaptors. $n = 3$ replicates per condition, $N = 3$. Bar graphs represent mean ± SD. Source data are provided as a Source data file.

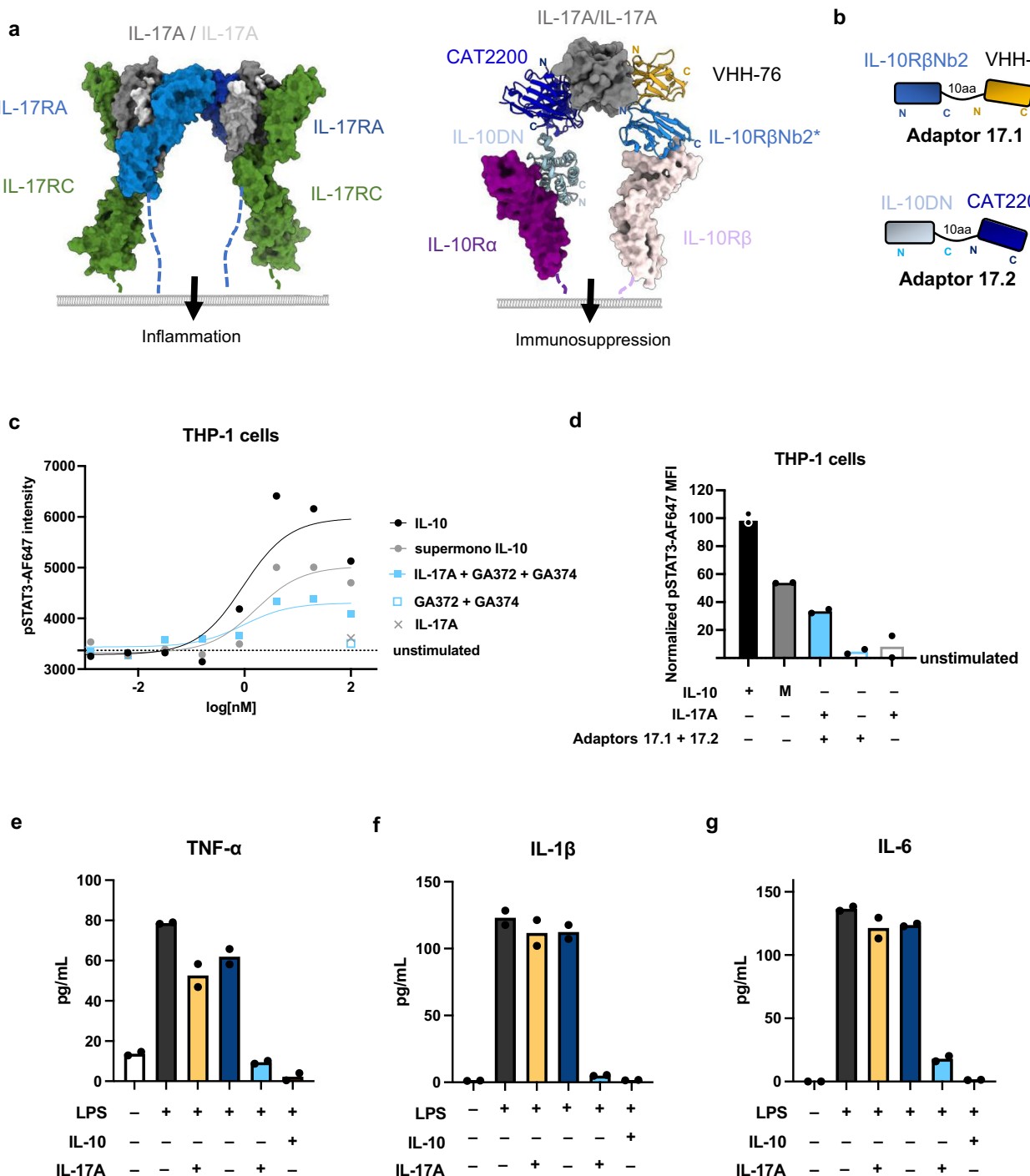

**Fig. 5 | IL-17→IL-10 cytokine adaptors mimic effects of IL-10. a** Model comparing endogenous IL-17A signaling through IL-17RA and IL-17RC (left, PDB ID: 7UWN) vs. signaling through IL-17→IL-10 cytokine adaptors (right) which dimerize IL-10Rα and IL-10Rβ. Model of IL-17→IL-10 adaptors is comprised of IL-17A binders CAT2200 (PBD ID: 2VXS) and netakimab fragment VHH-76 (PDB ID: 8B7W) overlaid with structures of IL-10 receptor complex (PDB ID: 6X93). **b** Cartoon representations of IL-17→IL-10 cytokine adaptor designs with 10 amino acid (aa) linker between modules. **c**, **d** IL-17→IL-10 adaptors induce a dose-dependent pSTAT3 signal in THP-1 cells. **c** Dose–response curves for phospho-STAT3 in THP-1 cell line stimulated for 20 min with human IL-10, a supermonomeric version of IL-10 or IL-17A with equimolar IL-17→IL-10 cytokine adaptors. Data represented as mean, $n = 2$, $N = 3$. **d** Normalized pSTAT3 response in THP-1 cells stimulated with 10 nM WT IL-10, human IL-10, a supermonomeric version of IL-10 (M), IL-17A, or IL-17A with the IL-17→IL-10 adaptors. Data are plotted as mean, $n = 2$, $N = 3$. **e**–**g** IL-17→IL-10 adaptors suppress LPS-induced production of inflammatory cytokines. TNF-α (**e**), IL-6 (**f**), and IL-1β (**g**) quantified by ELISA in supernatants from human PBMCs stimulated with 24 h with LPS and 10 nM IL-10, IL-17, or IL-17→IL-10 adaptors. Bar graphs represent mean, n = 2. Source data are provided as a Source data file.

molecules in mouse models. Although the adaptors display specific signaling and function in vitro, their activity in vivo will depend on local levels of cytokine expressed in diseased and normal tissue. Since target cytokines are also expressed systemically, dosing studies in animal disease models will be required to determine the therapeutic window for each adaptor. Thus, further studies are needed to determine the systemic effects of cytokine adaptors to assess safety and efficacy in animal models.

## Methods

### Ethical statement
All research in this study complies with ethical regulations related to the use of human samples. Samples obtained from the Stanford Blood Bank, approved by Stanford University Institutional Review Board under protocol number 13942.

### Materials
A list of materials is provided in the Supplementary Information (Supplementary Table 5).

### Mammalian cell lines and culture conditions
For cytokine adaptor protein expression, Expi293F cells (Gibco) were cultured in Expi293 expression media (Gibco) at 37 °C with 5% $CO_2$.

Signaling assays were conducted in YT-1 cells cultured in complete RPMI medium containing RPMI 1640-glutaMAX (Gibco) containing 10% FBS (Fisher Scientific), 4 mM GlutaMAX (Gibco), 25 mM HEPES (Gibco), 2 mM pyruvate (Gibco), non-essential amino acids (Gibco), and penicillin-streptomycin (Gibco) at 37 °C with 5% $CO_2$. THP-1 cells were cultured in complete RPMI medium containing beta-mercaptoethanol (Thermo Fisher).

### Human primary cells and culture conditions
Human peripheral mononuclear cells (PBMCs) were obtained from Stanford Blood Bank and cultured in complete RPMI at 37 °C with 5% $CO_2$.

### Cytokine adaptor design
TGF-β→IL-2 adaptors were designed by linking N-terminal VHH against IL-2Rβ or γc[34], to C-terminal scFv GC1008 against TGF-β[31].

IL-10→IL-2 adaptors were designed by linking N-terminal VHH against IL-2Rβ or γc[34] to C-terminal scFv 9D7[42] against IL-10 or extracellular domain of IL-10Rα.

IL-23→IL-10 adaptors were designed by linking VHH against IL-10Rβ (IL-10RβNb2) to VHH 22E11[59] against IL-23p35, and VHH 37D5[59] against IL-23p19 to a dominant negative, monomeric mutant of IL-10 with IL-10Rα.

IL-17→IL-10 adaptors were designed by linking VHH against IL-10Rβ (IL-10RβNb2) to the VHH-76 fragment of netakimab[60] against IL-17A, and CAT2200[53] against IL-17A to a dominant negative, monomeric mutant of IL-10 with IL-10Rα.

All structural figures were made using ChimeraX[61]. The structure of IL-10RβNb2 was predicted using ColabFold[62,63].

### Protein production and purification
For the expression of cytokine adaptors, the constructs described above were cloned into the pD649 mammalian expression vector containing C-terminal 6xHis-tags (ATUM). Expi-293F cells (Thermo Fisher, Cat# A14528) were transiently transfected using Expifectamine (Thermo Fisher) as per manufacturer instructions. Supernatant was harvested 3–5 days post transfection, incubated with Ni-NTA resin (QIAGEN) and eluted with 0.2 M imidazole followed by size-exclusion chromatography (SEC) on a Superdex 200 column (GE).

For the expression of cytokines used in signaling experiments, IL-10 was similarly expressed in Expi-293F cells[30]. For expression of IL-23, the p19 subunit and p40 subunit with a C-terminal His tag were cloned into separate pD649 vectors[29]. Expi-293F cells were co-transfected with p19 and p40 DNA, and protein was harvested 4 days post transfection. IL-23 was purified by Ni-NTA resin (QIAGEN) followed by SEC as described previously.

### Signaling assays in YT-1 cells, THP-1 cells, and human PBMCs
For dose–response signaling assays in YT-1 cells (Cellosaurus, Cat# CVCL_EJ05) and THP-1 cells (ATCC, Cat# TIB-202), cells were plated in a 96-well plate at 200,000 cells per well in RPMI. TGF-β (R&D), IL-23 (R&D), or IL-10 were prepared at the indicated concentrations with equimolar cytokine adaptors in RPMI complete media. The cytokine mix was added to stimulate YT-1 or THP-1 cells for 20 min at 37 °C. Cells were fixed with 1.5% paraformaldehyde for 10 min at room temperature, then permeabilized with 100% methanol for 30 min at −20 °C. Intracellular STAT activation was assayed by staining with Alexa Fluor® 488 or 647 conjugated anti-STAT5 (pY694) or anti-STAT3 (pY705) (BD) for 1 h at room temperature at a 1:100 dilution. Mean fluorescence intensities were measured using a CytoFlex flow cytometer (Beckman Coulter) and analyzed in Prism v9.3.0 (GraphPad).

For signaling assays in activated human T cells, human peripheral blood mononuclear cells (PBMCs) were isolated from samples from the Stanford Blood Center and cryopreserved. Human T cells were activated in a 12-well plate pre-coated with 2 µg/mL anti-human CD3 clone OKT3 (BioLegend) in complete RPMI containing 5 µg/mL soluble anti-CD28 (BioLegend) and cultured at 37 °C with 5% $CO_2$ for 48 h. Activated cells were rested overnight without stimulation in RPMI, stained with anti-human CD8 (BioLegend) and anti-human CD4 (BioLegend) for 15 min at 4 °C at a 1:100 dilution. Cells were then plated in a 96-well plate (200,000 cells per well) and stimulated with TGF-β, IL-2, IL-17, IL-23, IL-10, with or without equimolar cytokine adaptors for 20 min at 37 °C. Cells were fixed, permeabilized, and assayed for pSTAT or pSMAD activity by flow cytometry (Supplementary Fig. 4).

### Proliferation assays in human PBMCs
For TGF-β adaptor proliferation assays, human T cells were activated for 48 h with anti-CD3 (BioLegend) and anti-CD28 (BioLegend) as described above, rested overnight in RPMI, and then cultured for 6 days with 500 ng/mL TGF-β (R&D) with equimolar cytokine adaptors, or IL-2. On day 6, cells were harvested and divided in half for immediate analysis and further stimulation for intracellular cytokine staining. For analysis, cells were stained with human TruStain FcX (BioLegend), anti-CD8 PE (BioLegend), and anti-CD4 (FITC) at a 1:100 dilution and quantified by flow cytometry. For intracellular cytokine staining, cells were cultured with PMA and ionomycin, Golgiplug (BD) and Golgistop (BD) for 6 h at 37 °C. Cells were then fixed and permeabilized with BD fix/perm buffer (BD), and stained with anti-CD4, anti-CD8, anti-CD25, anti-TNFα, and anti-IFNγ, and assayed by flow cytometry. All antibodies used were purchased from BioLegend.

### LPS-mediated inflammation assay in human PBMCs
For LPS-mediated inflammation assays, human PBMCs were thawed after cryopreservation and rested at $2 \times 10^6$ cells/mL for 2 h in RPMI. Cells were then cultured for 24 h with 1 ng LPS in the presence or absence of 10 nM IL-10, IL-23, and cytokine adaptors. Supernatants were harvested and analyzed for IL-1β, IL-6, and TNF-α using IL-1β, IL-6, and TNF-α DuoSet ELISAs (R&D).

### Statistics and reproducibility
Differences between samples were tested by one-way non-parametric ANOVA (Kruskal–Wallis test) with multiple comparisons. All measurements were taken from distinct samples. Data were plotted and statistics calculated using Prism v. 9.3.0 (GraphPad). Data in dose–response curves and bar graphs are plotted as mean of technical

replicates ± standard deviation (SD). Technical replicates are denoted as *n*. Data presented are representative of three or more biologically independent experiments, denoted as *N*. Dose–response signaling curves are plotted with each data point representing the mean of $n = 2$ technical replicates, and each graph representative of at least $N = 3$ independent experiments. Sample size was determined based on similar studies published on cytokine signaling. Comparisons between groups were planned before statistical testing. No data were excluded from the analyses. The experiments were not randomized. Investigators were not blinded to allocation during experiments and outcome assessment.

## Reporting summary

Further information on research design is available in the Nature Portfolio Reporting Summary linked to this article.

## Data availability

The data generated in this study are provided in the Source data file provided with this paper. Protein structures used in this work are listed below. 2B55I, 2PJY, 6X93, 7S2R, 1LK3, 6XDQ, 7UWN, and 8B7W. Source data are provided with this paper.

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

## Acknowledgements

The authors would like to thank all members of the Garcia lab for helpful feedback and discussion. K.C.G. is supported by NIH-RO1-AI51321, Yosemite Innovation Fund, Ludwig Fund, and Howard Hughes Medical Institute (HHMI). G.C.A. is supported by the Hertz Foundation Fellowship and Stanford University Medical Scientist Training Program grant T32-GM007365. N.E.R. is supported by DP2 AI177915. C.B.B. is supported by the Arc Institute Fellowship.

## Author contributions
G.C.A., K.D.H., G.E.R., C.R.G., R.A.S., C.B.B., S.C.W., L.S., M.Y., C.H., V.G.P., and N.E.R. designed and conducted all experiments. C.R.G. and K.C.G. conceived the project. G.C.A. and K.C.G. wrote the manuscript with input from all authors.

## Competing interests
G.C.A., C.R.G., R.A.S., K.D.H., and K.C.G. are inventors on patent WO2023201206A1 on the cytokine adaptor technology described in this manuscript. The remaining authors declare no competing interests.
