## [Transparent Peer Review file · Nature Communications]

Redirecting immune signaling with cytokine adaptors

Corresponding Author: Professor K. Christopher Garcia

Version 0:

Reviewer comments:

Reviewer #2

(Remarks to the Author)

I appreciate the adaptations of this interesting paper presenting a novel concept in cytokine therapeutics for Nature Communications.

Reviewer #3

(Remarks to the Author)

This manuscript presents the potentially attractive idea of harnessing immunosuppressive cytokines (TGF-beta and IL-10) for eliciting a potentially immunostimulatory signal, and vice versa, using an inflammatory cytokine (IL-23) to exert an immunosuppressive signal. The work features the development and in vitro characterization of cytokine adaptors. These cytokine adaptors contain an 'input' part made of cytokine-binding molecules specific to a cytokine and an 'output' made of binding molecules specific to a dimeric output cytokine receptor. The authors developed TGF-beta-gated IL-2 (TGF-beta>IL-2), IL-10-gated IL-2 (IL-10>IL-2), and IL-23-gated IL-10 (IL-23>IL-10) cytokine adaptors. These cytokine adaptors were characterized in vitro by using the cell lines YT-1 and THP-1, human CD4+ and CD8+ T cells, and human PBMCs, as well as in LPS-stimulated human PBMCs.

Based on the accompanying point-by-point responses, this manuscript represents a revised version of the authors' work. Some of the previously raised points have been addressed. However, some concerns still remain, as detailed below.

1. The previously voiced concern on Emax and recommending the use of titrated concentrations of cytokine adaptors is still valid. Thus, the single-chain TGF-beta>IL-2 adaptors are supposed to increase the Emax compared to that seen with IL-2. However, the Emax values shown in Fig. 1e,f and Fig. 2c-f are difficult to reconcile with this hypothesis. The observed results are also not explained by a difference in IL-2Ralpha (CD25) surface abundance of YT-1 cells versus human primary T cells, as IL-2 by itself (i.e., without TCR stimulation) does not upregulate CD25 in human conventional T cells. If the authors claim to obtain different results, they should present them. Also, the authors should provide results using titrated concentrations of the different single-chain TGF-beta>IL-2R adaptors versus titrated concentrations of IL-2 in human primary CD4+ regulatory T cells (trimeric IL-2Rs), CD4+ conventional T cells (dimeric IL-2Rs, or trimeric IL-2Rs if activated by TCR), and NK cells (dimeric IL-2Rs).

2. The authors cite an article by Fox et al. published 1993 in Cellular Immunology, which showed that TGF-beta inhibited the proliferation of Con A-prestimulated, IL-2-stimulated human T cells in vitro. However, another publication by de Jong et al. published 1994 in International Immunology (doi: 10.1093/intimm/6.4.631) demonstrated that TGF-beta was able to activate human CD4+ T cells in vitro. Thus, the previously raised points is still valid. Are the data shown in Fig. 2h to be interpreted as the result of proliferation or mere survival?

3. I appreciate the authors' explanation about the discrepancy between pSTAT5 agonism and pSTAT3 antagonism, but I still do not understand how the same IL-10 molecule is able to induce – via the action of the IL-10>IL-2R adaptors – STAT5 phosphorylation at a concentration of 0.1 nM IL-10, while simultaneously achieving maximal pSTAT3. Should the increase in pSTAT5 not be paralleled by a concomitant decrease in pSTAT3 because the input cytokine is 'converted' to stimulate IL-2Rs instead of IL-10Rs?

Version 1:

Reviewer comments:

Reviewer #3

(Remarks to the Author)

I have no further comments.

Reviewer #2 (Remarks to the Author):

I appreciate the adaptations of this interesting paper presenting a novel concept in cytokine therapeutics for Nature Communications.

We thank the reviewer for their comments.

Reviewer #3 (Remarks to the Author):

This manuscript presents the potentially attractive idea of harnessing immunosuppressive cytokines (TGF-beta and IL-10) for eliciting a potentially immunostimulatory signal, and vice versa, using an inflammatory cytokine (IL-23) to exert an immunosuppressive signal. The work features the development and in vitro characterization of cytokine adaptors. These cytokine adaptors contain an 'input' part made of cytokine-binding molecules specific to a cytokine and an 'output' made of binding molecules specific to a dimeric output cytokine receptor. The authors developed TGF-beta-gated IL-2 (TGF-beta>IL-2), IL-10-gated IL-2 (IL-10>IL-2), and IL-23-gated IL-10 (IL-23>IL-10) cytokine adaptors. These cytokine adaptors were characterized in vitro by using the cell lines YT-1 and THP-1, human CD4+ and CD8+ T cells, and human PBMCs, as well as in LPS-stimulated human PBMCs. Based on the accompanying point-by-point responses, this manuscript represents a revised version of the authors' work. Some of the previously raised points have been addressed. However, some concerns still remain, as detailed below.

We thank the reviewer for their comments. Each point is addressed below.

1. The previously voiced concern on Emax and recommending the use of titrated concentrations of cytokine adaptors is still valid. Thus, the single-chain TGF-beta>IL-2 adaptors are supposed to increase the Emax compared to that seen with IL-2. However, the Emax values shown in Fig. 1e,f and Fig. 2c-f are difficult to reconcile with this hypothesis. The observed results are also not explained by a difference in IL-2Ralpha (CD25) surface abundance of YT-1 cells versus human primary T cells, as IL-2 by itself (i.e., without TCR stimulation) does not upregulate CD25 in human conventional T cells. If the authors claim to obtain different results, they should present them.

The signaling assays in Fig. 1e-f were conducted in YT-1 cells (no IL-2Ra expression), whereas the signaling assays in Fig 2c-f were conducted in human primary T cells pre-activated with plate-bound anti-CD3, soluble anti-CD28, and IL-2 (high IL-2Ra expression). A head-to-head comparison of the split chain adaptors T.1 and T.2 vs. single chain adaptor T.3 is best seen in Fig 2e-f. Here, our main claim is that the single chain adaptor T.3 has an improved Emax and left shifted EC50 compared to the pair of T.1 and T.2. The reason for the discrepancy in the EC50 of the IL-2 signaling curves between Fig. 1e and Fig. 2e-f is that the pre-activated primary T cells express high levels of IL-2Ralpha (CD25). While the EC50 of IL-2 signaling undergoes a left-shift in cell-types expressing IL-2Ra that confers sensitivity to IL-2 (which is CD25-dependent), the adaptors do not bind to IL-2Ra, so the EC50 of the

adaptors is not affected by IL-2Ra expression. We have conducted IL-2Ra staining on human PBMCs before and after plate activation to demonstrate the upregulation of IL-2Ra (Extended Fig. 2).

Also, the authors should provide results using titrated concentrations of the different single-chain TGF-beta>IL-2R adaptors versus titrated concentrations of IL-2 in human primary CD4+ regulatory T cells (trimeric IL-2Rs), CD4+ conventional T cells (dimeric IL-2Rs, or trimeric IL-2Rs if activated by TCR), and NK cells (dimeric IL-2Rs).

We understand the reviewer's interest in seeing the adaptor activity on different cell subsets, but the engineering technology we show is agnostic to cell type: TGFb will trigger signaling by the adaptors on any cells that express IL-2Rβ and gamma-C. Unlike natural IL-2, the adaptors are not CD25 dependent so we anticipate that differences in IL-2Rβ and gamma-C expression levels will affect responses in different cell types. We also anticipate that different cell types will exhibit different response thresholds to the adaptors based on their intrinsic wiring (e.g. Glassman et al. eLife 2021), but we think this is beyond the intended scope of the paper. In the current version of the manuscript, signaling is shown in YT-1 cells, a human NK cell line that expresses IL-2Rb and IL-2Rg (**Fig 1e-f**) as well as CD8+ and CD4+ conventional primary T cells pre-activated with anti-CD3, anti-CD28 that express IL-2Rb, gamma-C, and IL-2Ra (**Fig. 2e-f**).

2. The authors cite an article by Fox et al. published 1993 in Cellular Immunology, which showed that TGF-beta inhibited the proliferation of Con A-prestimulated, IL-2-stimulated human T cells in vitro. However, another publication by de Jong et al. published 1994 in International Immunology (doi: 10.1093/intimm/6.4.631) demonstrated that TGF-beta was able to activate human CD4+ T cells in vitro. Thus, the previously raised points is still valid. Are the data shown in Fig. 2h to be interpreted as the result of proliferation or mere survival?

In our hands, TGFβ is uniformly immunosuppressive on B and T cells through inhibition of proliferation. We have not seen evidence for T cell activation. This is consistent with the vast majority of published literature and is the rationale for inhibiting TGFb for cancer immunotherapy. Indeed, in our lab, targeting TGFβ to B or T cells causes cell type specific inhibition of proliferation. As one example, Tgfbr2 knockout enhances the proliferation of CD8+ T cells in tumors, as shown by Ki67 staining in Sun et al, Nature Immunology 2023 below:

(Fig. S5I, doi: 10.1126/sciimmunol.adh1306)

3. I appreciate the authors' explanation about the discrepancy between pSTAT5 agonism and pSTAT3 antagonism, but I still do not understand how the same IL-10 molecule is able to induce – via the action of the IL-10>IL-2R adaptors – STAT5 phosphorylation at a concentration of 0.1 nM IL-10, while simultaneously achieving maximal pSTAT3. Should the increase in pSTAT5 not be paralleled by a concomitant decrease in pSTAT3 because the input cytokine is 'converted' to stimulate IL-2Rs instead of IL-10Rs?

The assay in Fig. 3e was conducted with the concentration of IL-10 held constant at 10nM. The X-axis indicates the concentration of IL-10>IL-2R adaptor. Thus, when 0.1nM IL-10>IL-2R adaptor is mixed with 10nM IL-10, this is akin to ~0.1nM IL-2 receptor agonist, which can induce robust STAT5 phosphorylation. However, the IL-10-induced STAT3 phosphorylation is not minimized because there is still 100X the concentration of IL-10 present compared to the IL-10 inhibitor (10nM IL-10 vs. 0.1nM inhibitor). The relative affinities of the IL-10 binding modules and the IL-2 receptor binding modules will further influence the relative pSTAT5 agonism vs. pSTAT3 antagonism. In short, an increase in pSTAT5 need not be paralleled by concomitant decrease in pSTAT3 at 0.1nM because there is excess IL-10 in the assay capable of signaling.

REVIEWERS' COMMENTS

Reviewer #3 (Remarks to the Author):

I have no further comments.

We thank the reviewer for their comments.